# Long-term health-related quality of life, healthcare utilisation and back-to-work activities in intensive care unit survivors: Prospective confirmatory study from the Frisian aftercare cohort

Lise F. E. Beumeler[1,2]*, Anja van Wieren[2], Hanneke Buter[2], Tim van Zutphen[1,3], Gerjan J. Navis[3], E. Christiaan Boerma[2]

1 Campus Fryslân, University of Groningen, Leeuwarden, The Netherlands, 2 Department of Intensive Care, Medical Centre Leeuwarden, Leeuwarden, The Netherlands, 3 Faculty of Medical Sciences, University Medical Centre Groningen, Groningen, The Netherlands

* l.f.e.beumeler@rug.nl

## Abstract

### Purpose

More substantial information on recovery after Intensive Care Unit (ICU) admission is urgently needed. In a previous retrospective study, the proportion of non-recovery patients was 44%. The aim of this prospective follow-up study was to evaluate changes in Health-Related Quality of Life (HRQoL) in the first year after ICU-admission.

### Methods

Long-stay adult ICU-patients ($\geq$ 48 hours) were included. HRQoL was evaluated with the Dutch translation of the RAND-36 item Health Survey (RAND-36) at baseline via proxy measurement, and at three, six, and twelve months after ICU admission. Subsequently, the relation between physical functioning, healthcare utilisation, and work activities was explored.

### Results

A total of 81 patients were included in this study. Fifty-five percent of patients did not meet criteria for full recovery and were allocated to the Non Recovery (NR)-group (Physical Functioning domain-score: 35 [15–55]). Baseline physical HRQoL differed significantly between the Recovery (R) and NR-group. Patients in the NR-group received home care more often and had higher healthcare utilisation (44 versus 17% in the first three months post-ICU, p = 0.013). Only fourteen percent of NR-patients were able to participate in work activities. Moreover, NR-patients persistently showed impaired overall HRQoL throughout the year after critical illness.

**Data Availability Statement:** All data are available from the Zenodo database (10.5281/zenodo. 6656128).

**Funding:** This research received no specific grant from any funding agency in the public, commercial, or not-for-profit sectors.

**Competing interests:** All authors declare that they have no conflict of interest.

## Conclusions

Limited recovery in ICU survivors is reflected in overall impaired HRQoL, as well as in far-reaching consequences for patients' healthcare needs and their ability to reintegrate into society. In our study, baseline HRQoL appeared to be an important predictor of long-term outcomes, but not Clinical Frailty Scale (CFS) score. And, (proxy-derived) HRQoL may help to identify patients at risk of long-term non-recovery.

## Introduction

The primary aim of Intensive Care Unit (ICU) treatment is to improve the chance of survival for critically ill patients. Over the last few decades, enhanced treatment options and advanced technologies have resulted in an increased number of ICU survivors [1–3]. Despite this success, evaluation of patient-centred outcomes, commonly assessed by HRQoL scores, has revealed substantial proportions of survivors experiencing persistent physical, mental, and cognitive health problems [3–5]. Previous research in this population has indicated a large proportion of ICU patients suffer from long-term limitations in physical functioning in the first year after admission [6]. Therefore, using physical functioning as a marker for recovery in HRQoL may provide researchers and clinicians with more information on overall vulnerability in the post-ICU period.

Although critical care research has embraced the need for more robust information regarding ICU recovery, long-term follow-up of ICU patients has been burdened with high loss to follow-up, heterogeneity of results, and a lack of uniform methodology. Further, assessing the impact of critical illness is complicated in the acute setting, in particular due to the lack of baseline information regarding preadmission health status. Additionally, there is a need for more evidence regarding the impact of long-term health problems on the ability to participate in work activities, as well as on healthcare utilisation. Information regarding healthcare utilisation is often predominantly focussed on the amount of hospital and ICU readmissions [7]. However, more in-depth information regarding the use of physical therapy, dietary consultations, and home care, among others, is limited.

In this study, the primary aim was to prospectively confirm percentages of non-recovery (NR) patients at twelve months after ICU admission. Additionally, we aimed to obtain a highly detailed follow-up, including baseline HRQoL, the effect of NR on health-care utilisation and back-to-work activities.

## Material and methods

### Study design and population

This prospective, single-centre, observational study was performed in a tertiary teaching hospital with a mixed ICU, located in Leeuwarden, the Netherlands. The ICU is an 18-bed mixed medical-surgical unit that admits close to 1500 patients a year, with close to half of these admissions following elective surgery [8].

All adult patients admitted to the ICU between May 20 and November 27 of 2019, with a length of stay (LOS) ICU of $\geq$ 48 hours, who were able to read and understand the Dutch language, were included in this study. The cut-off value on LOS ICU of 48 hours was used as a large proportion of patients in this ICU ward with a shorter LOS are admitted per protocol and will be discharged within two days without mayor complications. It is commonly known

that these patients have a lower risk of long-term health problems. Sample size was based on the average number of long-stay patients admitted to the local ICU ward in six months due to the explorative nature of this study. The follow-up measurements were conducted throughout the first year after ICU admission. Participating patients who did not survive until the one year follow-up, did not complete the end-of-study HRQoL measurement, or were lost to follow-up were excluded from analysis.

This study, including a deferred consent procedure for patients that were not able to provide consent at baseline, has been evaluated and approved by the local research ethics committee of the Medical Centre Leeuwarden (Regionale Toetsingscommissie Patiëntgebonden Onderzoek, Leeuwarden, The Netherlands; METC-number: RTPO 1055). The study protocol was registered online (ClinicalTrials.gov identifier: NCT04154995). All patients provided written informed consent. A deferred consent procedure was instated to make sure baseline measurements could be performed if the patient was unable to give consent due to, for instance, sedation or delirium. When clinical evaluation showed that the patient was able to give an informed response, official consent was obtained. As a consequence, patients with severe cognitive problems after awakening, e.g. postanoxic coma, or with inevitable ICU mortality were excluded.

## Data collection

At baseline (LOS ICU ≤72h), a proxy of the patient was asked to complete the RAND-36, in order to evaluate the patient's HRQoL prior to ICU admission [9]. This questionnaire, which is very similar to the Medical Outcome Study Short-Form-36 (MOS SF-36), consists of nine domains, as described in the previously conducted retrospective study [10]. Higher scores indicate better subjective health status. Patients were asked to complete the RAND-36 again at three, six and twelve months after ICU admission. In line with the previously applied identification method, all patients with a physical functioning (PF) domain score below age-adjusted reference value -based on a Dutch healthy control of 65–75 years old group at twelve months-were allocated to the physical NR-group [6, 11]. Patients with higher scores were assigned to the recovery (R) group. Additional survey information about work activities and healthcare utilisation were obtained. In case of non-response, patients were reminded via e-mail or telephone. When a hospital visit deemed not to be feasible, a researcher visited the patient at their home or rehabilitation environment. If applicable, the questionnaire was completed verbally with the assistance of a researcher.

Baseline and ICU characteristics were collected as standard care and retrieved from electronic patient data files. Medical comorbidities were indicated as stated in the National Intensive Care Evaluation [12]. The clinical frailty scale (CFS), consisting of one domain with a score range of one for 'Very fit' to nine for 'Terminally ill', was used to evaluate pre-admission physical performance and independence [13].

## Statistical analysis

After study completion, data were processed in a coded file in January 2021. After taking into account the sample size of the study, variables were summarised as median [interquartile range, IQR] and frequencies (percentage). Differences between the R-group and NR-group, both at baseline and during the first year after admission, were assessed per predefined protocol using appropriate statistical tests. P-values were estimated by using the Mann-Whitney U test, the two-sided Fisher's Exact test in case of dichotomous data, or the Pearson Chi-Squared test in case of categorical variables within more than 2 groups. Repeated measures in the R-group and NR-group were tested using Friedman's test. In case of statistical significance, post-

hoc analysis using a Wilcoxon signed-rank test was conducted with a Bonferroni correction. A two-sided p-value <0.05 was considered as statistically significant, or p<0.008 after Bonferroni correction. SPSS Statistics for Windows, Version 27 (IBM) and GraphPad Prism version 5.0.4. for Windows (Graphpad Software) were used for statistical testing and visualisation of the data. RAND-36 domain score outcomes were displayed visually using Microsoft Excel (Microsoft Corporation). Results of this study were reported using the Strengthening the Reporting of Observational Studies (STROBE) checklist [10].

### Bias and missing data

Missing data due to either early mortality, severe cognitive impairments, or other reasons for loss to follow-up, are common in critical care research, and might be associated with disease burden and lack of recovery. To limit potential bias, baseline characteristics of the study population and patients of which no completed RAND-36 could be obtained at the end of the study, were compared and reported in the S1 Table.

## Results

### Patient selection and group allocation

Between May and November 2019, 107 patients were screened for eligibility, of whom 81 patients gave informed consent for this study. At twelve months, 65 patients completed the RAND-36 questionnaire and were consequently included in the analysis (Fig 1). Thirty-six patients (55%) were allocated to the NR-group with a median physical functioning (PF)-

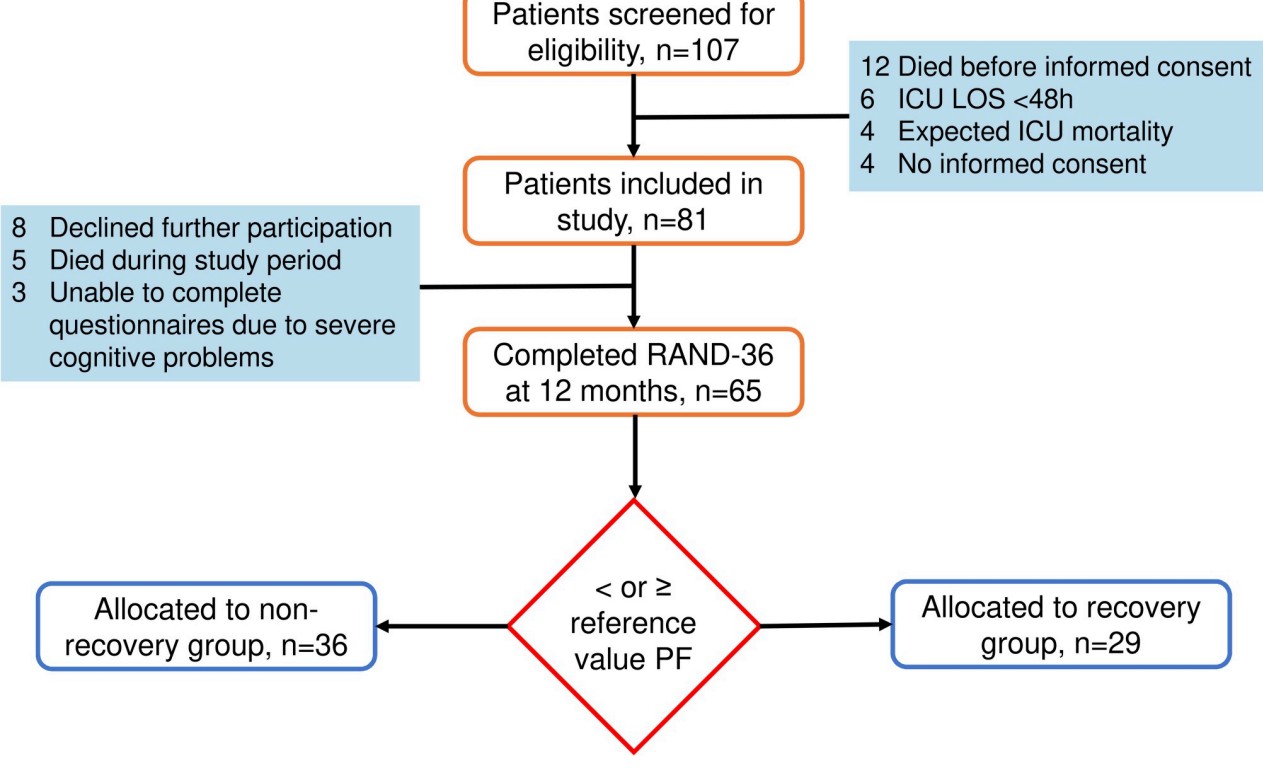

**Fig 1. Flowchart study inclusion and group allocation.**

domain score at twelve months of 35 [15–55]. Twenty-nine patients (45%) were added to the R-group (median PF-domain score at twelve months: 95 [83–95]).

## Comparison of group characteristics

In this study population, 25% was female and the median age of the group was 66 [57–74] years. Patients were managing well in daily life prior to admission (CFS: 3 [2–4]). The majority of admission types was medical (52%) and the median LOS ICU was 5 [4–10] days (Table 1). Patients in the physical NR-group were older, were more frail, had a higher LOS ICU, and were in need of mechanical ventilation for a longer period of time.

## Baseline HRQoL and frailty

In the three months before ICU-admission, patients allocated to the NR-group scored significantly lower on physical HRQoL domains (Physical functioning, Role physical, Energy/ Fatigue, Bodily pain) and general health perception (Table 1). In addition, these patients were more frail before ICU-admission (CFS 2 versus 3, with score 2 indicating 'Well/Fit', i.e. no active disease symptoms, and score 3 indicating 'Managing well', i.e. medical problems are well controlled). There were no differences in mental HRQoL or health change subscale scores.

## Physical functioning at baseline and at three, six, and twelve months

In the R-group, a statistically significant increase in the PF-domain score was observed over time ($\chi 2$ = 8.424, p = 0.038). Post-hoc analyses indicated a significant difference after correcting for multiple testing in PF-domain scores when comparing physical functioning at both three and six months with scores at twelve months (p = 0.006 and p = 0.004, respectively). In contrast, domain scores in the NR-group remained unaltered ($\chi 2$ = 7.284, p = 0.063). Between-group analysis revealed that PF-domain scores were significantly higher for the R-group at baseline, at three, and at six months after ICU-admission (p<0.001) (Fig 2).

## Healthcare utilisation and work participation

Overall, there was no difference either in the number of ICU readmissions within the year after discharge or in rehabilitation intensity between the R and NR-group (Table 2A). Patients made use of six appointments with healthcare professionals (HCP) during the first three months after discharge (Table 2B). During this period of time, patients in the NR-group received home care more often (44 versus 17%, p = 0.013). This difference in the percentage of people receiving home care persisted in the later periods (three to six and nine to twelve months after admission). It was only during the last term that the NR-group made more use of HCP (5 [1–12] over 2 [0–3], p = 0.004).

Throughout the first year after ICU-admission, close to a third of the patients in both groups worked less hours than before ICU-admission (Table 3). Before ICU admission more than half of patients participated in work activities, with a median total of 4 [0–26] hours of work per week. Shortly after admission, work participation dropped to 15%. More than a third of patients actively participating in work activities did so for less hours than before ICU-admission. After twelve months, work participation was at 30%. Although there was no significant difference in the amount of patients participating in work activities before ICU admission, the R-group did work more hours per week (p = 0.04). However, at three, six, and twelve months after discharge, the NR-group consistently had a lower number of patients who were able to work (p = 0.004, p = 0.006, p = 0.001, resp.).

**Table 1. Comparison of group characteristics at baseline and after ICU admission.**

| Characteristics | All | NR at 12m | R at 12m | p-value |
|---|---|---|---|---|
| | N = 65 | N = 36 (55%) | N = 29 (45%) | |
| **Pre-ICU** | | | | |
| Frailty (CFS) (1–9) | 3 [2–4] | 3 [3–4] | 2 [1–3] | **0.001** |
| Baseline HRQoL (0–100) | | | | |
| Physical functioning | 65 [45–95] | 55 [29–81] | 95 [65–100] | <**0.001** |
| Social functioning | 88 [63–100] | 88 [50–100] | 88 [63–100] | 0.132 |
| Role physical | 50 [0–100] | 25 [0–81] | 100 [25–100] | **0.011** |
| Role emotional | 100 [67–100] | 100 [67–100] | 100 [83–100] | 0.602 |
| Mental health | 88 [68–96] | 82 [67–93] | 88 [72–96] | 0.204 |
| Energy/Fatigue | 70 [40–80] | 58 [35–75] | 70 [60–85] | **0.017** |
| Bodily pain | 78 [55–100] | 71 [43–100] | 88 [68–100] | **0.049** |
| General health perception | 60 [44–85] | 50 [30–75] | 70 [50–90] | **0.003** |
| Health change | 50 [25–50] | 25 [19–50] | 50 [25–50] | 0.19 |
| **Demographic factors** | | | | |
| Female, n (%) | 16 (25) | 12 (33) | 4 (14) | 0.087 |
| Age | 66 [57–74] | 71 [62–77] | 63 [50–73] | **0.038** |
| BMI (kg/m2) | 27 [24–31] | 28 [24–31] | 27 [23–29] | 0.172 |
| APACHE III | 76 [58–97] | 79 [61–94] | 75 [55–99] | 0.428 |
| **Comorbidities** | | | | |
| Malignancy, n (%) | 6 (9) | 3 (8) | 3 (10) | 1.000 |
| Diabetes, n (%) | 11 (17) | 9 (25) | 2 (7) | 0.094 |
| COPD, n (%) | 8 (12) | 7 (19) | 1 (3) | 0.066 |
| CVA, n (%) | 6 (9) | 3 (8) | 3 (10) | 1.000 |
| CKD, n (%) | 6 (9) | 5 (14) | 1 (3) | 0.213 |
| Multicomorbidity[a], n (%) | 7 (11) | 6 (17) | 1 (3) | 0.120 |
| Psychiatric history, n (%) | 15 (23) | 10 (28) | 5 (17) | 0.384 |
| **Aetiology** | | | | |
| Admission, n (%) | | | | |
| Medical | 34 (52) | 16 (44) | 18 (62) | 0.326 |
| Elective surgical | 16 (25) | 11 (31) | 5 (17) | |
| Acute surgical | 15 (23) | 9 (25) | 6 (21) | |
| Sepsis, n (%) | 14 (22) | 11 (31) | 3 (10) | 0.069 |
| CPR, n (%) | 10 (15) | 3 (8) | 7 (24) | 0.096 |
| Delirium, n (%) | 24 (37) | 13 (36) | 11 (38) | 1.000 |
| **ICU morbidity** | | | | |
| LOS ICU | 5 [4–10] | 7 [4–15] | 4 [3–8] | **0.006** |
| Mechanical ventilation (days) | 3 [1–6] | 4 [2–10] | 2 [1–4] | **0.009** |
| Renal replacement therapy | 11 (17) | 7 (19) | 4 (14) | 0.742 |

Abbreviations: BMI, Body Mass Index; APACHE, Acute Physiology and Chronic Health Evaluation; CFS, Clinical Frailty Scale; COPD, Chronic Obstructive Pulmonary Disease; CVA, Cerebrovascular Accident; CKD, Chronic Kidney Disease; CPR, Cardiopulmonary Resuscitation; LOS, Length of Stay; ICU, Intensive Care Unit
[a]multicomorbidity was indicated as at least two medical comorbidities at baseline

## Overall health-related quality of life at baseline and at three, six, and twelve months

When visualised by use of a spider web chart, overall HRQoL revealed marked differences between groups at all time points, including baseline (Fig 3). This difference in HRQoL at

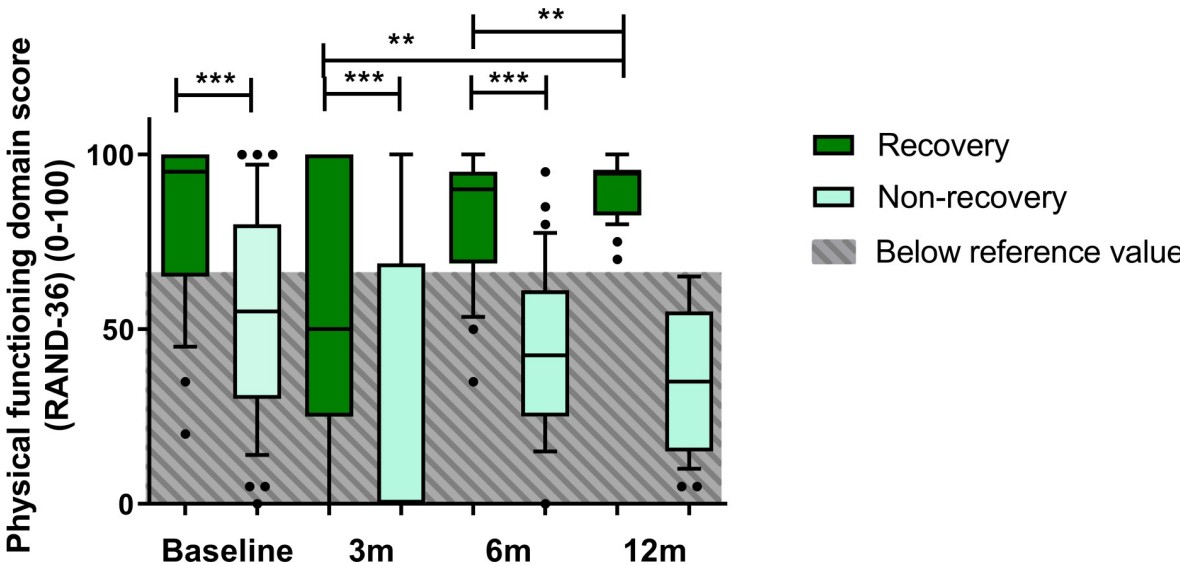

**Fig 2. Course of physical functioning domain scores during the first year after ICU-admission for R-group and NR-group in boxplot with 10-90th percentile whiskers.** ** p≤0.01 *** p<0.001.

baseline was not reflected by a clinically relevant difference in frailty scores (Table 1). Comparing role limitations due to physical problems (RP) in both groups resulted in the highest score difference at twelve months after admission (NR: 0 [0–50]; R: 100 [75–100], p<0.001). Mental health and Role Emotional domain scores remained high in both groups. The full data on domain scores at baseline, three, six and twelve months can be found in the S2 Table.

**Table 2. Post-ICU characteristics (A) and healthcare utilisation (B) over the first year after ICU admission.**

| A. Post-ICU characteristics | All | NR at 12m | R at 12m | p-value |
|---|---|---|---|---|
| | N = 65 | N = 36 (55%) | N = 29 (45%) | |
| ICU readmission in 1 year, n (%) | 2 (3) | 2 (6) | 0 (0) | 0.498 |
| Rehabilitation intensity | | | | |
| No rehabilitation | 12 (19) | 5 (14) | 7 (24) | 0.372 |
| Self-initiated or primary care | 17 (26) | 9 (25) | 8 (28) | |
| Cardiac rehabilitation programme | 20 (31) | 10 (28) | 10 (35) | |
| General rehabilitation centre | 14 (22) | 10 (28) | 4 (14) | |
| Nursing home | 2 (3) | 2 (6) | 0 (0) | |
| **B. Healthcare utilisation (over a 12 week period)** | | | | |
| **Discharge to 3 months post-ICU** | | | | |
| Number of appointments HCP[a] | 6 [3–15] | 6 [2–16] | 6 [3–11] | 0.830 |
| Received home care, n (%) | 21 (32) | 16 (44) | 5 (17) | **0.013** |
| **3–6 months post-ICU** | | | | |
| Number of appointments HCP[a] | 3 [1–11] | 3 [1–11] | 3 [1–9] | 0.613 |
| Received home care, n (%) | 18 (28) | 15 (42) | 3 (10) | **0.012** |
| **9–12 months post-ICU** | | | | |
| Number of appointments HCP[a] | 3 [1–8] | 5 [1–12] | 2 [0–3] | **0.004** |
| Received home care, n (%) | 16 (25) | 13 (36) | 3 (10) | **0.021** |

[a] Healthcare professional (HCP): General practitioner, Medical specialist, Social worker, Physical therapist, Occupational therapist, Speech therapist, Dietician, Alternative medicine, Psychological help, Company doctor

**Table 3. Participation in paid and volunteer work activities and hours worked over the first year after ICU admission.**

| Work activities | All | NR at 12m | R at 12m | p-value |
|---|---|---|---|---|
| | N = 65 | N = 36 (55%) | N = 29 (45%) | |
| **Before ICU admission** | | | | |
| Participates in work activities, n (%) | 36 (55) | 17 (47) | 19 (66) | 0.297 |
| Total hours of work, h/w | 4 [0–26] | 2 [0–10] | 20 [0–36] | **0.040** |
| **3 months after ICU discharge** | | | | |
| Participates in work activities, n (%) | 10 (15) | 1 (3) | 9 (31) | **0.004** |
| Works less hours than before ICU-admission, n (%) | 23 (35) | 12 (33) | 11 (38) | 0.782 |
| **6 months after ICU discharge** | | | | |
| Participates in work activities, n (%) | 15 (23) | 4 (11) | 11 (38) | **0.006** |
| Works less hours than before ICU-admission, n (%) | 21 (32) | 11 (31) | 10 (35) | 0.778 |
| **12 months after ICU discharge** | | | | |
| Participates in work activities, n (%) | 21 (32) | 5 (14) | 16 (55) | **0.001** |
| Works less hours than before ICU-admission, n (%) | 22 (34) | 14 (39) | 8 (28) | 0.283 |

## Discussion

In this prospective twelve-month observational period, more than half of the long-term ICU survivors showed no significant sign of physical recovery. These results substantiate the findings of our previously published retrospective study on recovery in an ICU outpatient clinic cohort [6]. In addition, after ICU-admission this was associated with shortcomings in self-efficacy and societal participation. Persistent physical NR was a marker for impairment in (almost) all domains of HRQoL. Ultimately, proxy-derived HRQoL at baseline helped to identify patients at risk for non-recovery.

It is commonly known that age, pre-admission health status, and frailty impact recovery after critical illness. Although there was an imbalance in age between groups in our study, we believe it is unlikely to be the sole cause of the observed lack of recovery. At first glance, CFS at baseline did not differ up to the point of clinical risk identification, despite there being a statistically significant difference. However, a closer look at baseline HRQoL revealed marked differences between groups. Specifically in the physical HRQoL domains, patients that did not recover after 12 months experienced more health-related impairments before admission. These findings may represent the rehabilitation potential of this patient group. However, as these patients also had a higher LOS in ICU and more days on mechanical ventilation and

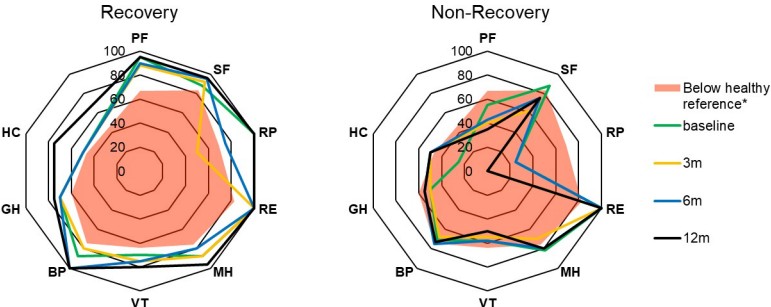

**Fig 3. Health-related quality of life in R- and NR-group at baseline, three, six, and twelve months after admission, with visual indication of healthy reference value per domain of the RAND-36 questionnaire.** Abbreviations: PF, Physical Functioning; SF, Social Functioning; RP, Role Physical; RE, Role Emotional; MH, Mental Health; VT, Vitality; BP, Bodily Pain; GH, General Health perception; HC, Health Change.

there was only a modest difference in frailty before admission, contributing the lack of recovery solely to pre-admission health seems inappropriate. Nevertheless, proxy-derived HRQoL may help to perform an adequate risk assessment for non-recovery, and could potentially be simplified by an isolated PF-domain score in the acute hospital setting. In addition, a subsequent assessment at three months after discharge has added value in the identification of long-term non-recovery, and can be a trigger for further rehabilitation. This adds to the existing literature, since pre-ICU data on quality of life and physical functioning comes from studies in the elective surgical group [14, 15].

In general, our findings are consistent with previously reported impaired recovery of physical functioning and HRQoL in ICU survivors. Firstly, in a study by Hofhuis et al. (2021), physical functioning domain scores varied between 6 and 59 from baseline to 10 years after ICU-discharge. This patient cohort never reached the age-adjusted reference value for the physical functioning domain. Secondly, in a follow-up study investigating physical functioning between three and twelve months after ICU discharge, researchers observed that the physical component score (PCS) remained far below age-adjusted reference values [16]. However, some differences need to be addressed. In a follow-up study conducted in 156 post-ICU patients, severity of illness was associated with physical recovery at the six months follow-up [17]. Interestingly, in our data set, patients with physical non-recovery did not have higher severity of illness scores. This dissimilarity may be due to group allocation based on physical recovery rather than ICU-characteristics. In conclusion, despite current rehabilitation options, critical illness survivors demonstrate long-term non-recovery in physical functioning.

Due to these long-term health problems, ICU survivors require more healthcare during and after hospital discharge compared to non-ICU patients, the latter reflected in a higher number of emergency room visits and hospital readmissions [7]. In a recent Dutch cohort study, ICU survivors were found to have up to five times higher healthcare costs compared to a healthy control group [18]. Our study shows that physical NR-patients may contribute more to this extreme increase in costs over the first year after admission with a primary focus on the need for home care and assistance in daily living. Targeting this patient group in future interventions may have a positive impact on healthcare costs.

Furthermore, the inability to return-to-work of ICU survivors is one of the most prevalent personal and social consequences of a long-term ICU admission. In a recent systematic review and meta-analysis of 52 studies, roughly one-third of the ICU survivors that were employed prior to ICU admission were jobless up to five years after ICU admission [19]. A prospective study in the north of the Netherlands indicated that the work rate (percentage of full-time) of a long-stay ICU-cohort was only 32.2% at six months after ICU discharge [20]. As disturbing as these results already are for all ICU survivors, return-to-work in NR-patients seems to be even worse, as not even 15% of patients participate in work activities twelve months after admission. The dire situation of this specific group warrants more extensive and elaborate aftercare interventions to ensure that these people have a higher chance of societal reintegration and regain a sense of purpose.

This study provides valuable information regarding pre-ICU health status, with in-depth assessment of HRQoL before admission in the acute setting, and recovery after critical illness. Yet, our study is limited by the number of patients and the heterogeneous origin of ICU-admission. Despite the in-depth information provided due to the longitudinal follow up design with several time points, the findings represent a select patient group in the northern part of the Netherlands. Results may not be identical in a different, for example academic or international, setting. Furthermore, the number of lost-to-follow-up has the potential to create unaccounted bias, although the percentage is lower than reported in previous literature on post-ICU follow-up services and research [4]. Moreover, as this study has a longer and more

extensive follow-up than our regular specialised outpatient clinic, it is notable that the researchers managed to complete the follow up the majority of participants despite the ongoing COVID-19 pandemic. Group characteristics of these dropouts make it unlikely to contribute to a substantially lower percentage of non-recovery (S1 Table).

In conclusion, long-term recovery after critical illness is limited in a proportion of ICU survivors. This lack of recovery is reflected in overall impaired HRQoL and untenable consequences for patients' healthcare needs, as well as their ability to reintegrate in society. In our study, baseline HRQoL appeared to be an important predictor of long-term outcomes, but not CFS. And (proxy-derived) HRQoL may help to identify patients at risk of long-term non-recovery. It is essential to investigate rehabilitation potential of patients that are unable to recover within the current aftercare setting. Personalised preventative and aftercare interventions to support patients at risk are urgently needed.

## Supporting information

**S1 Table. Characteristics of lost-to-follow-up.**
(DOCX)

**S2 Table. HRQoL domain scores at baseline and at 3, 6, and 12m.**
(DOCX)

## Author Contributions

**Conceptualization:** Lise F. E. Beumeler, Anja van Wieren, Hanneke Buter, Tim van Zutphen, Gerjan J. Navis, E. Christiaan Boerma.

**Data curation:** Lise F. E. Beumeler, Anja van Wieren.

**Formal analysis:** Lise F. E. Beumeler, E. Christiaan Boerma.

**Investigation:** Lise F. E. Beumeler, Anja van Wieren, Tim van Zutphen, E. Christiaan Boerma.

**Methodology:** Lise F. E. Beumeler, Hanneke Buter, Tim van Zutphen, Gerjan J. Navis, E. Christiaan Boerma.

**Project administration:** Lise F. E. Beumeler, Anja van Wieren.

**Resources:** E. Christiaan Boerma.

**Supervision:** Hanneke Buter, Tim van Zutphen, Gerjan J. Navis, E. Christiaan Boerma.

**Validation:** Lise F. E. Beumeler, Gerjan J. Navis, E. Christiaan Boerma.

**Visualization:** Lise F. E. Beumeler, E. Christiaan Boerma.

**Writing – original draft:** Lise F. E. Beumeler, Hanneke Buter, Tim van Zutphen, Gerjan J. Navis, E. Christiaan Boerma.

**Writing – review & editing:** Lise F. E. Beumeler, Anja van Wieren, Hanneke Buter, Tim van Zutphen, Gerjan J. Navis, E. Christiaan Boerma.

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
