## [Decision Letter · Decision Letter 0]

19 May 2022

PONE-D-22-03052Long-term health-related quality of life, healthcare utilisation and back-to-work activities in Intensive Care Unit survivors: prospective confirmatory study from the Frisian Aftercare CohortPLOS ONE

Dear Dr. Beumeler,

Thank you for submitting your manuscript to PLOS ONE. After careful consideration, we feel that it has merit but does not fully meet PLOS ONE’s publication criteria as it currently stands. Therefore, we invite you to submit a revised version of the manuscript that addresses the points raised during the review process.

We look forward to receiving your revised manuscript.

Kind regards,

Ashham Mansur, MD, PhD

Academic Editor

PLOS ONE

Journal Requirements:

2. Please clarify in the ethics statement whether the METC has specifically approved the deferred consent procedure for patients unable to provide consent at the baseline.

Additional Editor Comments:

According to the reviewer comments, your manuscript needs an extensive revision, before it can be further considered for publication in plos one.

Reviewers' comments:

Reviewer's Responses to Questions

**Comments to the Author**

1. Is the manuscript technically sound, and do the data support the conclusions?

Reviewer #1: Partly

Reviewer #2: Yes

Reviewer #3: Partly

2. Has the statistical analysis been performed appropriately and rigorously? 

Reviewer #1: Yes

Reviewer #2: Yes

Reviewer #3: No

3. Have the authors made all data underlying the findings in their manuscript fully available?

Reviewer #1: Yes

Reviewer #2: Yes

Reviewer #3: No

4. Is the manuscript presented in an intelligible fashion and written in standard English?

Reviewer #1: Yes

Reviewer #2: Yes

Reviewer #3: Yes

5. Review Comments to the Author

Reviewer #1: I have a few comments:

1. “Long-stay adult patients (≥ 48 hours)”--this definition is inappropriate; >= 48 hours is not a prolonged stay.

2. The sample size is too small for a exploratory analysis.

3. "Patients in the NR-group received home care more often and had higher healthcare utilisation."--always show the quantity and statistical inference in result section.

4. The novelty of the study is unclear because there has been many such studies in the literature, though the journal may not put novelty as a priority.

5. "Missing data due to either early mortality"--this can cause the competing risk in the model.

6. You can also study the risk factors for poor functional recovery.

Reviewer #2: Thank you for the opportunity to review this manuscript concerning long-term follow up of critical illness survivors; an area of great importance. The manuscript describes a prospective, single-center, follow-up study. Congratulations to the authors on a well-performed and interesting study. It is impressive that you obtained full follow up on 65 out of 81 patients.

I have some comments and suggestions:

Abstract

In the last sentence under “purpose” you write: “In this prospective follow-up study, changes in Health-Related Quality of Life (HRQoL), twelve months after ICU admission, were observed in long-stay ICU patients.” This is actually a result. I would recommend that the last sentence in “purpose” is a brief description of the aim of the study, “The aim of this study was to….”

In “methods” you write “Long-stay adult patients” – I think you need to include that they are long-stay ICU-patients (although I know that there is a limited number of words in an abstract, it still needs to make sense, when you read only the abstract)

You write “the relation between non-recovery, healthcare utilisation, and work activities was explored”. I find this sentence confusing. What is non-recovery? (I can guess, but you have not introduced the term). Can you somehow rephrase so that it makes sense for someone, who has not met the term non-recovery previously? Patients, who survive, but somehow are not well anyway?

In the methods section you should mention which tool you use to assess HRQoL.

You need to explain abbreviations the first time you use them – you write NR (which I guess to be non-recovery) without explanation.

You write “untenable consequences” – I do not believe that consequences can be untenable, however, situations can.

Manuscript

Introduction: well described, clear aim. I would probably use an additional small paragraph to clarify why you focus so much on physical function, rather than psychological or cognitive, for example.

Material and methods

Study population: interesting that half of your patients are admitted following elective surgery. Could you state what kind of surgery it is? What do you think it means, in order of interpreting your results? I would guess that patients, who have elective surgery, are not quite as ill as for example chronically ill medical patients?

You write that “Participating patients who did not survive until the one year follow-up, did not complete the end-of-study HRQoL measurement, or were lost to follow-up were excluded from analysis”. This puzzles me. A patient, who survived 11 months and completed all follow up until then is not interesting? Could you explain the rationale for this?

Results

You should describe the different scales/measurements used, in the methods section. You here mention a Clinical Frailty Score (which I find really good) without having introduced it.

Healthcare utilisation and work participation

You write: ”Overall, there was no difference either in the number of ICU readmissions within the year after discharge or in rehabilitation intensity” – between the R and the NR group?

Page 9, line 188 – this can not be table 1 as well? Table 2, perhaps?

Discussion

You write: “Furthermore, the inability to return-to-work of ICU survivors is one of the most prevalent social and economic consequences of a long-term ICU admission”. Is it really? Quite a few of the patients are retired, and ICU patients are getting older and older. I agree that it is important, however, I believe that it is more important on a personal level, and that the costs associated with the highly increase healthcare needs are more prevalent.

Your ”Limitations” section is very brief. I would elaborate a bit more, for example on the high number of elective patients, the limited number of patients etc. However, I would also add a “Strengts” section, as your study does have several strengths. It can make sense to present strengths and limitations next to each other.

Conclusion

You write: “In conclusion, long-term recovery after critical illness is limited in ICU survivors.” As I see your study, the whole point is that recovery is limited in SOME survivors, and not (or, much less) in others, and that some cut-off value in a sub-score of the RAND-instrument seems promising in order to identify them?

Reviewer #3: This is a single centre cohort study of ICU survivors that explores a range of relevant patient centred outcomes. The sample size is rather small, with quite high levels of loss to follow-up. This is certainly a topical area, and there are interesting data included on employment status and visits to hospital that are often not reported in follow up ICU studies. I have the following comments:

1. In the abstract, I think it important to note that baseline (ie pre-ICU illness) HRQoL and health status was different between the R and NR groups for the reasons outlined below.

Methods

2. Population – what was the justification for the 48 hours cut-off for prolonged stay. Was this MV duration or any ICU LoS. This should be clarified.

3. What was the fate of those patients excluded based on cognitive impairment at the time of screening? How many were there? This could be a major source of inclusion bias especially as delirium and cognitive impairment are associated with adverse long terms outcomes.

4. The method of dichotomising the population should be justified more clearly in my view. Allocation of all patients below the age-adjusted reference value is potentially problematic as this presumably represents a population average? Did the authors consider a value more than a SD from the age adjusted reference as potentially more relevant given this is a population distribution of scores? They have in effect compared those in the lower versus highest 50% of the ‘normal’ population yet this distribution could be interpreted as including many people whose health is within the normal population? This seems a slightly odd way to classify a ‘non-responder’ especially.

5. In relation to point 4 (above) how was the baseline value for the RAND-36 used, given the known association between pre-existing measures of health and longer term HRQoL scores (in fact this is likely the main determinant of longer term HRQoL). For example this is shown in reference 13 and other larger cohort studies. The key issue with the method used is that the baseline differences are showing patients with different pre-existing health rather that the impact of critical illness alone.

Results

6. The number of patients screened was 107, but it is unclear whether these were all potentially eligible patients given there are 1500 admissions per year. This is a potential source of selection/inclusion bias. Can the authors clarify? The STROBE flow diagram in figure 1 does not clarify this. Surely there must have been many more than 107 patients requiring >48 hours during the study period?

7. For the patients classified as NR versus R groups, can the authors clarify whether this was based on individual age/gender matched predicted status, or a single population value? Assuming it is the former can more information be provided about where these data come from, eg is it based on the Dutch population?

8. The data in table 1 presents the comparison of those classified as recovered versus non-recovered. Ca the authors clarify several points:

a. Baseline HRQoL is dramatically different, especially in the physical functioning domain, energy, and general health domains. This confirms previous studies suggesting that pre-existing health is probably the major determinant of post ICU HRQoL or physical functioning status (see point 5 above). This is a non-modifiable predictor, but it effectively means that the non-recovered patients were likely ‘non-recovered’ to some extend BEFORE their critical illness, ie this is not about recovery, rather pre-existing health status. Can the authors comment and consider this.

b. The methods state that Bonferroni correction for the many tests was used but this table does not indicate whether this has been applied or which are considered significant after application.

c. For co-morbidities it is important to clarify and state which tool was used to capture these as it will likely influence the prevalence and range reported.

9. The observation in the test of a trend towards improvement in the R group but not in the NR group is another indicator that this may simply reflect the pre-illness health status. In many ways it is the relationship to recalled baseline health within individuals that is most relevant, for example percent of baseline. Could the authors evaluate that measure which may be more relevant? The non-response in the NR group likely reflects strongly pre-existing health trajectory and status?

10. Table 2 seems to be mislabelled as table 1 (page 9)

11. For the Use of appointments with HCPs, it is relevant to understand if these were scheduled or unscheduled care. The lack of differences would be expected if this was scheduled appointments? Can the authors provide more information about this measure?

12. In table 3, it is unclear what (voluntary) refers to. Is this voluntary work or does this intend to capture employment? Given there is a marked difference in age between the R and NR groups is this relevant to work, as I assume older people would be less likely to engage in work of any type?

13. The HRQoL data (page 11 and figure 3) illustrates the difficulty in interpreting the attributable impact of ICU admission versus differences in health status and health trajectory between the R and NR groups that pre-dated illness. The authors need to highlight this carefully and note that their approach to dichotomising the population may simply be identifying people on better versus poorer health trajectories at the time the critical illness required ICU admission. This has been demonstrated in previous work and is a key issue with understanding and interpreting ICU outcomes.

Discussion

14. The discussion would benefit from some re-structuring as the points made seem to ‘jump about’ a little. In my view the authors should focus far more on the difficulty in adjusting for pre-existing health in ICU outcomes studies. They have clearly shown that the baseline HRQoL based on relative judgement is a major determinant. I think the use of NR is rather misleading in this regard as this may simply represent different levels of pre-existing health. This has been illustrated in several similar larger cohort studies, with a key challenge being that baseline HRQoL is often not available due to the unscheduled nature of an ICU admission. This is actually a strength of this study. In my view the authors may not have fully considered the possible explanations for their findings. Their assumption that more rehabilitation may benefit NRs based on using baseline HRQoL is not really justified, as these people may not have capacity to recover?

6. PLOS authors have the option to publish the peer review history of their article (what does this mean?). If published, this will include your full peer review and any attached files.

Reviewer #1: **Yes: **Zhongheng Zhang

Reviewer #2: **Yes: **Helene Korvenius Nedergaard

Reviewer #3: **Yes: **Timothy Walsh

---

## [Author Response · Author response to Decision Letter 0]

27 Jun 2022

Dear academic editor and PLOS ONE reviewers,

The authors would like to thank you for your extensive and constructive comments regarding our submitted manuscript. The manuscript and additional files have been edited to address your concerns.

Please find the revised manuscript and related documents enclosed in this resubmission. We believe that your contribution has made our manuscript of higher quality and we thank you again for your careful consideration. 

We hope that our modifications render our manuscript in its current for suitable for publication in PLOS ONE.

Yours sincerely,

Lise Beumeler

 

Academic editor’s response:

We thank the editor for directing us to the style guides of PLOS ONE. We have adjusted the files accordingly.

2. Please clarify in the ethics statement whether the METC has specifically approved the deferred consent procedure for patients unable to provide consent at the baseline.

We have added information to our ethical statement regarding the METC approval for our deferred consent procedure.

The authors thank the editor for directing us towards the data availability information of PLOS ONE. After careful reconsideration, we conclude it is feasible to share the used data without causing privacy related or ethical issues. We have therefore made the used data available in the Zenodo database (10.5281/zenodo.6656128). 

Additional Editor Comments:

According to the reviewer comments, your manuscript needs an extensive revision, before it can be further considered for publication in plos one.

The authors thank the editor for the opportunity to improve our manuscript according to the expert feedback of the reviewers. We have improved the manuscript accordingly and feel confident that the revised manuscript will be a valuable addition to the content of PLOS ONE.

In the section below the authors’ response is described in detail when applicable. Changes made will be added in the revised manuscript.

Comments to Reviewer #1: 

I have a few comments:

1. “Long-stay adult patients (≥ 48 hours)”—this definition is inappropriate; >= 48 hours is not a prolonged stay.

The authors thank reviewer 1 for questioning our definition of a prolonged stay. We agree with the reviewer that giving a clear cut off value of prolonged stay of ICU patients can be complicated. The length of stay of our patient group is highly dependent of for example admission category (acute versus elective) and severity of illness. Nevertheless, for research purposes it is essential to identify a patient group at risk of long term health problems. Our local ICU population consists of a high number of elective surgical patients who will be admitted to the ICU per protocol for a maximum of 2 days, if there are no serious complications. It is commonly known that these patients have a lower chance of long term impairments after ICU admission and therefore we aimed to focus on patients with a length of stay ICU above 2 days. And indeed, our data show that this selection of patients was able to detect prolongued ICU stay with a median LOS ICU of 5 days with an IQR of 4-10. 

2. The sample size is too small for a exploratory analysis.

We thank the reviewer for the critical note on our sample size. We agree that we studied a modest amount of patients over time. Nevertheless, we believe that smaller-scale studies can provide us with valuable information regarding the recovery of individual patients, providing us with more delicate information on an n=1 level. With this study, we attempt to move forward from looking into individual predictors in large retrospective cohort studies to find targets for an individualized intervention program. We believe this prospective long-term study provides us with more in-depth information to achieve this goal. Nevertheless, we will take the remark into account when developing future studies in this patient group.

3. "Patients in the NR-group received home care more often and had higher healthcare utilisation."--always show the quantity and statistical inference in result section.

The authors thank the reviewer for pointing out the missing quantity and statistical inference of this result in our abstract. We adjusted this section accordingly. 

4. The novelty of the study is unclear because there has been many such studies in the literature, though the journal may not put novelty as a priority.

We thank the reviewer for being critical about the novelty of our research output. We agree with the reviewer that there are several studies reporting patient-reported outcomes, like health-related quality of life and healthcare utilisation. However, to our understanding we are the first group using physical functioning scores to allocate patients to a more frail/at risk group (Recovery/Non-recovery). Combining this classification with the in-depth change is HRQoL and healthcare utilisation gives a clear indication of the impact of non-recovery over time. In addition, we have completed a more extensive follow up than previously reported studies (ref 3, 4) with 3 follow up meetings and added proxy measurements to obtain more detailed information on quality of life before ICU-admission. Therefore we believe this study adds valuable new information to the research field. 

5. "Missing data due to either early mortality"--this can cause the competing risk in the model.

We agree with reviewer 1 that missing data due to early mortality can have a negative impact on the quality of the used model. As stated in the ‘Method’ section, we limit this potential bias by comparing the baseline characteristics of the patients included in the analysis with those of the patients we lost to follow-up. Despite a difference in sex and the incidence of chronic kidney disease these groups were identical. 

6. You can also study the risk factors for poor functional recovery.

We agree with reviewer 1 that studying the potential risk factors for poor functional recovery is an interesting idea to investigate. In a previous retrospective study, we have identified physical performance at the three month outpatient visit as a predictor for lack of recovery in physical functioning after one year, but this was not the case for commonly used characteristics for disease severity. The current manuscript is a prospective continuation of this previously conducted research: 

7. Beumeler LFE, van Wieren A, Buter H, van Zutphen T, Bruins NA, de Jager CM et al. Patient-reported physical functioning is limited in almost half of critical illness survivors 1-year after ICU-admission: A retrospective single-centre study. PloS one. 2020;15(12):e0243981.

Comments to Reviewer #2: 

Thank you for the opportunity to review this manuscript concerning long-term follow up of critical illness survivors; an area of great importance. The manuscript describes a prospective, single-center, follow-up study. Congratulations to the authors on a well-performed and interesting study. It is impressive that you obtained full follow up on 65 out of 81 patients.

The authors thank reviewer 2 for the compliment regarding our research. We agree that it encompasses a topic of great importance and are thankful for the contribution of reviewer 2 to further improve this manuscript.

I have some comments and suggestions:

Abstract

In the last sentence under “purpose” you write: “In this prospective follow-up study, changes in Health-Related Quality of Life (HRQoL), twelve months after ICU admission, were observed in long-stay ICU patients.” This is actually a result. I would recommend that the last sentence in “purpose” is a brief description of the aim of the study, “The aim of this study was to….”

The authors thank reviewer 2 for the suggestion to clarify our research aim in the abstract. We have adjusted the aim accordingly.

In “methods” you write “Long-stay adult patients” – I think you need to include that they are long-stay ICU-patients (although I know that there is a limited number of words in an abstract, it still needs to make sense, when you read only the abstract)

We have changed the patient definition to long-stay adult ICU-patients. Thank you for pointing this out.

You write “the relation between non-recovery, healthcare utilisation, and work activities was explored”. I find this sentence confusing. What is non-recovery? (I can guess, but you have not introduced the term). Can you somehow rephrase so that it makes sense for someone, who has not met the term non-recovery previously? Patients, who survive, but somehow are not well anyway?

The authors thank reviewer 2 for suggesting to use a different, more clear, term for non-recovery in the abstract. We have changed non-recovery to physical functioning, as this variable is used to allocate patients to either the recovery or the non-recovery group. We believe this makes this sentence more clear to readers that are not familiar with the term non-recovery.

In the methods section you should mention which tool you use to assess HRQoL.

We have added the tool we use to assess HRQoL in the method section of the abstract.

You need to explain abbreviations the first time you use them – you write NR (which I guess to be non-recovery) without explanation.

We have added the full description of the NR abbreviation to the results-section of the abstract. 

You write “untenable consequences” – I do not believe that consequences can be untenable, however, situations can.

The authors thank reviewer 2 for their feedback on the use of the word ‘untenable’. We have changed the abstract accordingly.

Manuscript

Introduction: well described, clear aim. I would probably use an additional small paragraph to clarify why you focus so much on physical function, rather than psychological or cognitive, for example.

We thank the reviewer for requesting additional rationale for our focus on physical functioning. In this manuscript, we further investigate the impact of physical functioning prospectively as a follow-up of a previously conducted retrospective analysis. In this study we identified a large proportion of our patients lack recovery in physical functioning throughout the year. We have added a clarification in the introduction section.

Material and methods

Study population: interesting that half of your patients are admitted following elective surgery. Could you state what kind of surgery it is? What do you think it means, in order of interpreting your results? I would guess that patients, who have elective surgery, are not quite as ill as for example chronically ill medical patients?

We thank the reviewer for this question and clarify this finding with pleasure. As can be read in table 1, 25% of our patients had been admitted after elective surgery, of which a large proportion of a cardiac nature (23% after acute surgery and 52% non-surgical). In fact, the majority of patients admitted to our ICU are patients undergoing elective surgery and staying in our ICU for 1 or 2 days per protocol. Using the 48 hour cut-off in our inclusion criteria aims to exclude these per-protocol visitors, as it is commonly known that these, indeed, are not quite as ill and most probably have less long-term health problems. However, when a patient has a longer length of stay after an elective surgery, this is predominantly caused by complications during or after the procedure, rendering them more vulnerable for a prolonged stay and long-term health problems. 

You write that “Participating patients who did not survive until the one year follow-up, did not complete the end-of-study HRQoL measurement, or were lost to follow-up were excluded from analysis”. This puzzles me. A patient, who survived 11 months and completed all follow up until then is not interesting? Could you explain the rationale for this?

The authors thank the reviewer for sharing their thoughts with us. As we used the HRQoL measurement at the 12 month follow up period to allocate patients to one of the two groups, it was not possible to include patients that were lost to follow up before the endpoint. These patients are most certainly interesting, which is why we investigated the baseline differences between the patients that completed the full follow-up and the patients that were unable to do so. There were minor differences between these groups at baseline (sex and incidence of chronic kidney disease), indicating that these groups were quite similar at baseline. Nevertheless, we lose some data regarding their recovery up until the moment of drop out. Other studies have ‘solved’ this issue by scoring all deceased patients with a score of 0 on the RAND-36 subscales, but we believe this does not adequately represent reality. In the future, we hope to also include these patients by using a mixed model which allows for some missing values. 

Results

You should describe the different scales/measurements used, in the methods section. You here mention a Clinical Frailty Score (which I find really good) without having introduced it.

We have added the description of the clinical frailty score in the method section.

Healthcare utilisation and work participation

You write: ”Overall, there was no difference either in the number of ICU readmissions within the year after discharge or in rehabilitation intensity” – between the R and the NR group?

We have added the group description to the first sentence of this paragraph.

Page 9, line 188 – this can not be table 1 as well? Table 2, perhaps?

We have changed the table number to table 2.

Discussion

You write: “Furthermore, the inability to return-to-work of ICU survivors is one of the most prevalent social and economic consequences of a long-term ICU admission”. Is it really? Quite a few of the patients are retired, and ICU patients are getting older and older. I agree that it is important, however, I believe that it is more important on a personal level, and that the costs associated with the highly increase healthcare needs are more prevalent.

The authors thank the reviewer for engaging in this Interesting discussion. As a large proportion of our patient group is above the retirement age, we agree that the effect of the inability to return to work will most probably be more evident on a personal and social level. We have adjusted this sentence accordingly. However, we want to point out that in the overall group there is also a significant amount of patients unable to fully return to work after discharge. With a median age of 67, this means that there is a high probability that not only the retired people are affected. An overall percentage of 34% that works less hours than before ICU-admission may imply still have an economic impact for the patient and their family, co-workers, etc.

Your ”Limitations” section is very brief. I would elaborate a bit more, for example on the high number of elective patients, the limited number of patients etc. However, I would also add a “Strengts” section, as your study does have several strengths. It can make sense to present strengths and limitations next to each other.

We have elaborated more extensively in the limitation paragraph of the discussion, adding some strengths of our research.

Conclusion

You write: “In conclusion, long-term recovery after critical illness is limited in ICU survivors.” As I see your study, the whole point is that recovery is limited in SOME survivors, and not (or, much less) in others, and that some cut-off value in a sub-score of the RAND-instrument seems promising in order to identify them?

We thank the reviewer for sharing their vision of the main conclusion of this manuscript. We adjusted the first sentence of the conclusion accordingly. We agree that one of the findings is also that the sub-score can be used for identification as was described in our previous retrospective study on this subject. However, in this manuscript we aim to investigate the implications when it comes to overall HRQoL and healthcare use in the post-ICU period.

Comments to Reviewer #3: 

This is a single centre cohort study of ICU survivors that explores a range of relevant patient centred outcomes. The sample size is rather small, with quite high levels of loss to follow-up. This is certainly a topical area, and there are interesting data included on employment status and visits to hospital that are often not reported in follow up ICU studies. I have the following comments:

1. In the abstract, I think it important to note that baseline (ie pre-ICU illness) HRQoL and health status was different between the R and NR groups for the reasons outlined below.

The authors thank reviewer 3 for their feedback on focussing more on the baseline differences in HRQoL of the R and NR groups. We have therefore added this to the result section of the abstract and addressed this issue further throughout the manuscript.

Methods

2. Population – what was the justification for the 48 hours cut-off for prolonged stay. Was this MV duration or any ICU LoS. This should be clarified.

We have specified the rationale behind including patients with a LOS of 48 hours or more in our method section. For further elaboration on this, we would like to refer reviewer 3 to our response to question 1 of reviewer 1 and the first question on the method section of reviewer 2. 

3. What was the fate of those patients excluded based on cognitive impairment at the time of screening? How many were there? This could be a major source of inclusion bias especially as delirium and cognitive impairment are associated with adverse long terms outcomes.

The authors acknowledge that being critical of potential bias in our study selection is necessary and thank reviewer 3 for this. Exclusion before screening due to cognitive impairment only occurred when patients were unable to give deferred informed consent, which indicated severe cognitive problems. For example, this could be the case for patients with a severe postanoxic coma. Patients with mild cognitive problems or delirium were mostly able to give deferred informed consent. We do not believe this is a major source of inclusion bias, as of the total patients screened for this study, 2 were excluded due to severe cognitive impairments. To clarify the extent of cognitive problems that would lead to exclusion, we have added an example to the last sentence of the study design paragraph. 

4. The method of dichotomising the population should be justified more clearly in my view. Allocation of all patients below the age-adjusted reference value is potentially problematic as this presumably represents a population average? Did the authors consider a value more than a SD from the age adjusted reference as potentially more relevant given this is a population distribution of scores? They have in effect compared those in the lower versus highest 50% of the ‘normal’ population yet this distribution could be interpreted as including many people whose health is within the normal population? This seems a slightly odd way to classify a ‘non-responder’ especially.

The authors thank the reviewer for the opportunity to critically reflect on our method of dichotomising the population. We agree that dividing the patients in two groups leads to a rather strict separation of responders and non-responders. In our previously published paper in which we retrospectively investigated this cut off (ref 7), we used Dutch, age-matched, healthy controls. However, measures of subjective HRQoL often result in a large spread of results, leading to large SD. In the control group we have used that would mean using a cut off value of 40 out of 100 on the physical functioning subscale, which still indicates a high amount of physical impairments. In addition, we found this cut off value to result in a ‘non-responder’ group with a median PF-subscale score of 35 out of 100 at 12 months, indicating this cut-off results in a non-responder group with severely impaired physical functioning.

5. In relation to point 4 (above) how was the baseline value for the RAND-36 used, given the known association between pre-existing measures of health and longer term HRQoL scores (in fact this is likely the main determinant of longer term HRQoL). For example this is shown in reference 13 and other larger cohort studies. The key issue with the method used is that the baseline differences are showing patients with different pre-existing health rather that the impact of critical illness alone.

The authors agree with reviewer 3 that it is impossible to consider long-term recovery in HRQoL without taking into account the baseline/pre-ICU HRQoL. However, as a large proportion of our patients are admitted acutely and probably experienced some health problems related to this illness before admission, we believe this may affect the baseline HRQoL. In most of our patients admission to the ICU is not the starting point of their illness and it is commonly known that frailty before ICU affects long-term outcomes. However, as shown in table 1, preadmission frailty did not differ extremely between patients (CFS 2 indicating the patient is ‘Well’ compared to CFS 3 indicating the patient is ‘Managing well’). Furthermore, we believe using the healthy age-adjusted control value for measuring the ability of recovery still gives an indication of which patients are in need of additional aftercare, despite their previous health-related problems. Nevertheless, we agree that discussing the baseline differences in HRQoL greatly improves this manuscript.

Results

6. The number of patients screened was 107, but it is unclear whether these were all potentially eligible patients given there are 1500 admissions per year. This is a potential source of selection/inclusion bias. Can the authors clarify? The STROBE flow diagram in figure 1 does not clarify this. Surely there must have been many more than 107 patients requiring >48 hours during the study period?

As stated in the results section of the manuscript, screening took place between May and November of 2019. The reviewer is right to state that in 2019 there were around 1500 admissions to our local ICU. Of these admission, more than half are per protocol after elective surgery. These patients usually have a LOS of max. 48 hours after the procedure, leaving them out of the current study. In addition, the summer period usually results in a lower amount of elective surgeries in general. The number of patients screened was therefore the actual amount of patients eligible in that year for this study. This number is similar to the number of patients that are eligible for our standard care outpatient clinic (150-200/year), which uses 48 hours in need of mechanical ventilation as a cut-off value.

7. For the patients classified as NR versus R groups, can the authors clarify whether this was based on individual age/gender matched predicted status, or a single population value? Assuming it is the former can more information be provided about where these data come from, eg is it based on the Dutch population?

We thank reviewer 3 for asking clarification on the healthy reference value used. In the method section we state that an age-adjusted reference value was used. We clarified the characteristics of this control group (Dutch, aged 65-75) in the Data collection section.

8. The data in table 1 presents the comparison of those classified as recovered versus non-recovered. Ca the authors clarify several points:

a. Baseline HRQoL is dramatically different, especially in the physical functioning domain, energy, and general health domains. This confirms previous studies suggesting that pre-existing health is probably the major determinant of post ICU HRQoL or physical functioning status (see point 5 above). This is a non-modifiable predictor, but it effectively means that the non-recovered patients were likely ‘non-recovered’ to some extend BEFORE their critical illness, ie this is not about recovery, rather pre-existing health status. Can the authors comment and consider this.

The authors agree that the baseline differences in HRQoL should be more prominently discussed in this manuscript, hence we have added a separate paragraph on baseline HRQoL and frailty in the result section. As mentioned above, there are notable differences in the physical HRQoL domains and subjective general health perception, but differences in pre-admission frailty significant, but small. In addition, the NR-group also had a longer LOS in ICU and more days of mechanical ventilation, making it plausible that the lack of recovery is caused by a combination of pre-ICU health and ICU-admission. 

b. The methods state that Bonferroni correction for the many tests was used but this table does not indicate whether this has been applied or which are considered significant after application.

We applied a Bonferroni correction for the analysis in the paragraph ‘Physical functioning at baseline and at three, six, and twelve months’ for the post-hoc tests used to assess changes over time in physical functioning domain scores. As mentioned in the methods, a p-value<0.008 was considered significant here. We did not apply the correction in the univariate comparison of group characteristics. We have added a clarification in the paragraph on physical functioning at the different time points.

c. For co-morbidities it is important to clarify and state which tool was used to capture these as it will likely influence the prevalence and range reported.

We used the checklist of the National Intensive Care Evaluation (NICE) database for the indication of comorbidities at ICU-admission. This has been added to the method section of the manuscript for clarification.

9. The observation in the test of a trend towards improvement in the R group but not in the NR group is another indicator that this may simply reflect the pre-illness health status. In many ways it is the relationship to recalled baseline health within individuals that is most relevant, for example percent of baseline. Could the authors evaluate that measure which may be more relevant? The non-response in the NR group likely reflects strongly pre-existing health trajectory and status?

We thank the reviewer for suggesting to investigate the change in HRQoL over time rather than the ability to reach the average for healthy controls. We agree with the reviewer that pre-illness health status plays an important role in the ability to recover. However, we do not believe this is the only factor contributing to the R-group improving over time and the NR-group remaining at a stable low level of physical functioning. We are sure the reviewer also agrees with us that the ability to recover after critical illness, if any illness, is impacted by a variety of factors, including but not limited to health-status before ICU. The improvement of the R-group over time could also indicate this is a less heterogenous group for which the current rehabilitation options fit better. One other explanation that we didn’t investigate in this study could be found in lifestyle behaviour or social/personal/environmental contextual factors. We therefore do not believe the allocation of our patients to groups based on physical functioning is a foul proof method, but it does provide a clear indicator for patients in need for additional or more user-centred aftercare interventions. Although investigating change over time would most definitely provide valuable additional insights, we do not believe this will provide us with a marker for rehabilitation needs in this specific setting.

10. Table 2 seems to be mislabelled as table 1 (page 9)

We relabelled table 1 as table 2.

11. For the Use of appointments with HCPs, it is relevant to understand if these were scheduled or unscheduled care. The lack of differences would be expected if this was scheduled appointments? Can the authors provide more information about this measure?

The number of HCP includes scheduled and unscheduled care and was asked in retrospect in our used questionnaire. However, taking into account the nature of the appointments it most probably reflects scheduled care as only the medical specialist could be consulted in an acute setting of unscheduled care. The authors are curious to know why reviewer 3 indicates the lack of differences to be as expected.

12. In table 3, it is unclear what (voluntary) refers to. Is this voluntary work or does this intend to capture employment? Given there is a marked difference in age between the R and NR groups is this relevant to work, as I assume older people would be less likely to engage in work of any type?

We thank the reviewer for directing us to the unclear terminology used. We have changed the title of table 3 accordingly to make it clear that we mean paid and volunteer work. The option for volunteer work was added as we believe that this indicates an important factor of participation in society. Despite the NR-group being older, a large proportion still participates in work related activities. The inability to return to these activities can have important implications on a personal, social and economic level. 

13. The HRQoL data (page 11 and figure 3) illustrates the difficulty in interpreting the attributable impact of ICU admission versus differences in health status and health trajectory between the R and NR groups that pre-dated illness. The authors need to highlight this carefully and note that their approach to dichotomising the population may simply be identifying people on better versus poorer health trajectories at the time the critical illness required ICU admission. This has been demonstrated in previous work and is a key issue with understanding and interpreting ICU outcomes.

We thank the reviewer for emphasizing again the importance to discuss pre-ICU health in our manuscript. Throughout the paper we have made changes accordingly.

Discussion

14. The discussion would benefit from some re-structuring as the points made seem to ‘jump about’ a little. In my view the authors should focus far more on the difficulty in adjusting for pre-existing health in ICU outcomes studies. They have clearly shown that the baseline HRQoL based on relative judgement is a major determinant. I think the use of NR is rather misleading in this regard as this may simply represent different levels of pre-existing health. This has been illustrated in several similar larger cohort studies, with a key challenge being that baseline HRQoL is often not available due to the unscheduled nature of an ICU admission. This is actually a strength of this study. In my view the authors may not have fully considered the possible explanations for their findings. Their assumption that more rehabilitation may benefit NRs based on using baseline HRQoL is not really justified, as these people may not have capacity to recover?

We acknowledge your valuable comment in the revised structure of the discussion, the conclusion, and the abstract. We have changed the discussion accordingly and added additional reflections on the fruitful discussions with the reviewers. 

The authors thank all the reviewers for their contribution to the quality of this manuscript.

---

## [Decision Letter · Decision Letter 1]

8 Aug 2022

Long-term health-related quality of life, healthcare utilisation and back-to-work activities in intensive care unit survivors: prospective confirmatory study from the Frisian aftercare cohort

PONE-D-22-03052R1

Dear Dr. Beumeler,

We’re pleased to inform you that your manuscript has been judged scientifically suitable for publication and will be formally accepted for publication once it meets all outstanding technical requirements.

Kind regards,

Ashham Mansur, MD, PhD

Academic Editor

PLOS ONE

Additional Editor Comments (optional):

Reviewers' comments:

Reviewer's Responses to Questions

**Comments to the Author**

1. If the authors have adequately addressed your comments raised in a previous round of review and you feel that this manuscript is now acceptable for publication, you may indicate that here to bypass the “Comments to the Author” section, enter your conflict of interest statement in the “Confidential to Editor” section, and submit your "Accept" recommendation.

Reviewer #1: All comments have been addressed

Reviewer #2: All comments have been addressed

2. Is the manuscript technically sound, and do the data support the conclusions?

Reviewer #1: Yes

Reviewer #2: Yes

3. Has the statistical analysis been performed appropriately and rigorously? 

Reviewer #1: Yes

Reviewer #2: Yes

4. Have the authors made all data underlying the findings in their manuscript fully available?

Reviewer #1: Yes

Reviewer #2: Yes

5. Is the manuscript presented in an intelligible fashion and written in standard English?

Reviewer #1: Yes

Reviewer #2: Yes

6. Review Comments to the Author

Reviewer #1: The comments are well addressed in the previous version; I am satisfied with this version and this can be good

Reviewer #2: Thank you to the authors for carefully answering all my queries. I find the answers sufficient and I believe that the manuscript has been imporved. There are some inherent limitations to the study (first and foremost the small sample size), but that cannot be changed, and I find that the results are of interest anyway.

7. PLOS authors have the option to publish the peer review history of their article (what does this mean?). If published, this will include your full peer review and any attached files.

Reviewer #1: No

Reviewer #2: **Yes: **Helene Korvenius Nedergaard

---

## [Editor Report · Acceptance letter]

11 Aug 2022

PONE-D-22-03052R1 

Long-term health-related quality of life, healthcare utilisation and back-to-work activities in intensive care unit survivors: prospective confirmatory study from the Frisian aftercare cohort 

Dear Dr. Beumeler:

I'm pleased to inform you that your manuscript has been deemed suitable for publication in PLOS ONE. Congratulations! Your manuscript is now with our production department. 

Kind regards, 

on behalf of

Dr. Ashham Mansur 

Academic Editor

PLOS ONE